# Osteoporosis as the Female-Specific Risk Factor for Dynapenia in Elderly Patients with Type 2 Diabetes

**DOI:** 10.3390/jcm13164590

**Published:** 2024-08-06

**Authors:** Chieh-Hua Lu, Sheng-Chiang Su, Feng-Chih Kuo

**Affiliations:** Division of Endocrinology and Metabolism, Department of Internal Medicine, Tri-Service General Hospital, National Defense Medical Center, Taipei 114202, Taiwan; undeca2001@gmail.com (C.-H.L.); shiyuan71@yahoo.com.tw (S.-C.S.)

**Keywords:** osteoporosis, dynapenia, body mass index, type 2 diabetes mellitus, females

## Abstract

**Aims:** Dynapenia is a noteworthy health issue contributing to increased risk of falling, but its co-occurrence with osteoporosis in elderly individuals with type 2 diabetes mellitus (T2DM) has not been well explored. Therefore, this study aimed to establish the association between osteoporosis and dynapenia, focusing on T2DM females due to their high prevalence of osteoporosis and fragility. **Methods:** We conducted a cross-sectional study to recruit a total of 103 T2DM patients (43 males and 60 females), aged between 50 and 80 years with median 68.0 years. Dual-energy X-ray absorptiometry (DXA) and dominant hand grip strength measurements were performed to define body composition, osteoporosis, and dynapenia in a sex-specific manner. **Results:** Higher prevalence of dynapenia and dyna-osteoporosis was observed in female T2DM patients with a significantly positive correlation between osteoporosis and dynapenia even after adjustment of body mass index (BMI). By performing a multivariate logistic regression analysis, both BMI and osteoporosis were identified as risk predictors for the development of dynapenia in female T2DM patients with odds ratios (95% CIs) of 1.234 (1.029–1.480) and 4.883 (1.352–17.630), respectively. **Conclusions:** Our results point out there is high, female-specific co-occurrence of osteoporosis and dynapenia in T2DM patients. Moreover, having osteoporosis and increased BMI might boost the risk of dynapenia in elderly females with T2DM.

## 1. Introduction

The remarkably increased prevalence of diabetes in elderly patients has been a social and economic burden, as well as an important medical issue [1]. According to the 10th edition of the International Diabetes Federation (IDF) Diabetes Atlas, more than 10% of adults are diagnosed as diabetic globally in 2021, and the prevalence of diabetes will continue to expand rapidly in the future, with the majority being type 2 diabetes mellitus (T2DM) [2]. Given the fact that T2DM is a chronic inflammatory disease characterized by dysregulated glucose metabolism and insulin resistance, the elderly patients with T2DM will often encounter body weight gain, central fat accumulation [3], decreased muscle strength [4], with an increased risk of osteoporotic fragility [5]. Moreover, visceral adipose tissue commonly exhibits compositional changes characterized by adipocyte hypertrophy and macrophage infiltration, which contribute to the development of insulin resistance. Conversely, addressing adipose tissue dysfunction to reduce lipid accumulation within the pancreas could mitigate pancreatic lipotoxicity and serve as an effective target for diabetes prevention and control [6]. Therefore, clinicians should not merely manage the glucose stability in patients with T2DM but also need to notice their body weight status with maintenance of physical activity for preserving lean mass and muscle strength, as recommended in the consensus report of the American Diabetes Associations (ADA) and the European Association for the Study of Diabetes (EASD) [7].

Previous studies on patients with T2DM have disclosed that a loss of skeletal muscle mass with an accelerated decline in muscle strength and functional capacity could significantly occur in the early T2DM patients [8,9]. The presentation of an aging-related loss of muscle mass and function, so-called sarcopenia, was initially proposed by Rosenberg IH in 1988 [10]. In recent decades, sarcopenia has been recognized as a diabetic complication [11] and associated with incapacitation, dependence, increased risk of falling, and fractures [12,13]. On the other hand, dynapenia, defined as an aging-related decline in muscle strength without significant loss of muscle mass, was increasingly recognized as another important health issue due to its high associations with physical inactivity and mortality [14,15]. Although low grip strength was referred to as possible sarcopenia during screening [16], dynapenia was also considered as a disorder independent from sarcopenia [15]. Notably, a longitudinal follow-up study has been conducted to assess three parameters, lean mass, gait speed, and handgrip strength, for the risk of recurrent falling and fractures and demonstrated that only a low grip strength was independently associated with recurrent falling [17]. The report highlights that we should raise greater attention to the diagnosis of dynapenia. Particularly, it is even more important for patients with diabetes since they are vulnerable for falling and the prevalence rate of dynapenia seems to be higher than sarcopenia in patients with T2DM [18].

Muscle and bone are both crucial tissues for maintaining overall health and functionality. A continuous reciprocity of physicochemical interaction between muscle and bone can preserve their mass, enhance muscle strength, and improve fracture resistance. However, elderly diabetic patients often experience limited activity, obesity, and chronic inflammation, leading to increased fatty infiltration within muscle and bone marrow. These factors contribute to aging-related degenerations, such as osteoporosis and sarcopenia [19]. For avoiding falling-related fragility fractures in patients with diabetes, early identification of subjects who have osteoporosis and low muscle strength will be crucial since muscle strength could be possibly enhanced after adequate training [20]. Currently, there is limited information regarding the co-existence of osteoporosis and dynapenia in elderly patients with long-duration diabetes.

Therefore, we recruited patients with type 2 diabetes aged between 50 and 80 years to assess the associations of osteoporosis and dynapenia in a sex-specific manner. Dual-energy X-ray absorptiometry (DXA) scans of hip and spine were applied as the gold standard for diagnosis of osteoporosis [21]. Meanwhile, we also measured their body composition using DXA and evaluated the dominant hand maximal grip strength using digital dynamometry for defining dynapenia. Therefore, we demonstrated the existence of risk factors by simultaneously assessing patients of dynapenia and osteoporosis in diabetes.

## 2. Material and Methods

### 2.1. Research Population

A total of 103 T2DM patients (43 males and 60 females) with median age of 68.0 years old, median 10 years of diabetes, and median 7.4% of HbA1c were recruited in this study during regular outpatient department follow-up. A cross-sectional study based on a multidimensional assessment of primary care users that were 60 years or older was conducted. The sample size was calculated using the expected proportion of the prevalence of frailty, which was 6.8% in Taiwan [22]. A kappa coefficient of 0.6 with a 95% confidence interval was used to generate a conservative sample size estimated using the Win–Pepi application [23]. Our subjects were admitted as outpatients at the Department of Endocrinology and Metabolism, Department of Internal Medicine, Tri-Services General Hospital, National Defense Medical Center, Taipei, Taiwan. The Tri-Services General Hospital is the tertiary center that offers the highest level of specialized care. The criteria for inclusion in this cross-sectional study are as follows: T2DM patients aged 50–80 years old receiving regular treatment of oral hypoglycemic agents or/and injectable therapy of either insulin or glucagon-like peptide 1 receptor agonist under stable conditions (in the Appendix A). Individuals with current acute illness of malignancy, cerebrovascular accident, myocardial infarction, heart failure, renal failure, hepatic failure, psychiatric illness, or pregnancy were excluded. All participants signed the Tri-Services General Hospital Institutional Review Board approval of this study (TSGHIRB No. 2-108-05-052) before participating in this study and agreed to the use of their personal information and related confidentiality in the research.

### 2.2. Anthropometric Measurements

Weight and height were measured using a standard scale and a wall-mounted stadiometer, respectively, and the weight was recorded to an accuracy of 0.1 kg, and the height to an accuracy of 0.1 cm. The calculation equation of BMI was weight (kg) divided by the square of height (m^2^). Waist circumference measurements were taken at the level of the midpoint between the lower edge of the last rib and the iliac crest, and hip circumference measurements were taken at their widest point. Waist and hip circumference measurements were recorded to the nearest 0.1 cm, and the waist-to-hip ratio was calculated as waist circumference divided by hip circumference. Blood pressure was measured in the right arm after 5 min of sitting still, repeated after 1 min, and the mean value of blood pressure was used in the analysis.

### 2.3. Hand Grip Strength Measurement

The hand grip strength was measured using a digital hand dynamometer. Measurements were made on the dominant hand and recorded to the nearest 0.1 kg. The width of the dynamometer was adjusted to the width best-suited to each participant. All participants were asked to hold the dynamometer by the side, not against the body, and with the elbow bent at a 90-degree angle, then squeeze the dynamometer with maximal force. The test was performed twice and the largest one was used for analysis.

### 2.4. Dual Energy X-ray Absorptiometry

The DXA was performed following the guidelines of the International Association for Clinical Densitometry [24] to quantify bone mineral density (BMD) and T-scores in the lumbar spine (L1–L4) and bilateral femoral neck regions. According to the recommendations of the International Osteoporosis Foundation (IOF) Scientific Advisory Board, any T-score in the spine or femoral neck region between −1 and −2.5 is defined as osteopenia, with a T-score of −2.5 or lower indicating the presence of osteoporosis [21]. DXA is also used as a standard method for quantifying overall and local body composition, including fat mass, lean mass, and bone mineral content. The DXA was performed by a certified technician. The participant, wearing a cotton gown with no metal attachments, lay supine in the center of the scanning area, maintaining a neutral position. The type of DXA machine used was the Lunar Prodigy Advance enCORE 2011 equipped with DXA software (enCORE V13.60.033).

The diagnosis of sarcopenia was based on the 2019 consensus update by the Asian Working Group for Sarcopenia [16] using cutoffs for height-adjusted appendicular skeletal muscle mass measured by DXA, <7.0 kg/m^2^ in men and <5.4 kg/m^2^ in women; and cutoffs for low muscle strength as handgrip strength < 28 kg for men and <18 kg for women. Dynapenia was defined as subjects who had low muscle strength with maintenance of normal levels of height-adjusted muscle mass.

### 2.5. Measurement of Biochemical Variables

Venus blood samples were drawn in an 8 h fasted state. Biochemical data, including blood lipids, glucose, liver, and kidney function, were measured. Plasma glucose concentrations were determined using the glucose oxidase method on a Beckman Glucose Analyzer II (Beckman Instruments, Fullerton, CA, USA). Serum levels of low-density lipoprotein (LDL) cholesterol, triglycerides, alanine aminotransferase (ALT), and creatinine were assessed using a Beckman Synchron LX20 analyzer (LX20; Beckman Coulter, Brea, CA, USA). The latest biochemical data were collected from the medical records of the recent 3 months from the date of the anthropometric measurements, grip strength, and DXA measurements.

### 2.6. Statistical Analysis

The continuous variables were evaluated by the Mann–Whiney U-test and shown as median values with quartiles. The categorical variables were analyzed using a Chi-square test and presented as percentages. The correlations of dynapenia with various parameters were assessed using Spearman rank–order correlations in a sex-specific manner. To identify the sex-specific predictors for dynapenia, multivariable stepwise logistic regression was conducted. Statistical calculations were carried out using SPSS 22.0 for Windows (SPSS, Inc., Chicago, IL, USA). Two-sided *p* values less than 0.05 were considered statistically significant.

## 3. Results

The baseline characteristics, medical prescription, grip strength, and DXA-defined body composition of the 103 patients with T2DM (60 females and 43 males) had been described and compared in our previous report [25] and are further listed in the Appendix A. Generally, females presented higher peripheral fat distribution with lower grip strength, lower total and regional lean mass, and lower bone mass than males, without significant differences in age, years of diabetes, and glycemic control. In this study, we further calculated the height-adjusted appendicular skeletal muscle mass along with the dominant hand grip strength for diagnosis of sarcopenia and dynapenia [16]. The sex-specific prevalence rate of sarcopenia, dynapenia, osteopenia, osteoporosis, and their co-occurrence are presented in Table 1. Notably, T2DM females presented a significantly higher prevalence of dynapenia (36.7% in females versus 7% in males, *p* = 0.001), a non-significant trend of higher prevalence of osteoporosis (28.3% in females versus 16.3% in males, *p* = 0.154), and a meaningfully increased co-existence of dynapenia and osteoporosis (16.7% in females versus 2.3% in males, *p* = 0.020). This finding suggested there is a female-predominant association between the development of dynapenia and osteoporosis.

To explore which parameters are crucial in the development of dynapenia in T2DM females, we further performed an exploratory factor analysis to survey the associations among age, BMI, several representative parameters of body components (waist-to-hip ratio (WHR) and android-to-gynoid fat ratio represented the central fat accumulation; arms’ fat-to-lean ratio represented limb skeletal muscle fat infiltration; grip strength represented the muscle function; appendicular skeletal muscle divided by the square of height (ASM/ht^2^) referred to the adjusted skeletal muscle mass; lumbar BMD represented the bone mass) and disorders related to loss of skeletal muscle mass, bone mass and strength (sarcopenia, dynapenia, and osteoporosis) as shown in Table 2. Factor loadings higher than 0.5 were considered a stronger association between the variables in the group. Intriguingly, the factor analysis divided T2DM females into two groups. In one group, the rise in BMI is negatively correlated with sarcopenia but not with dynapenia (factor loadings of 0.979 in BMI, −0.645 in sarcopenia, and 0.257 in dynapenia). In the other group, the occurrence of osteoporosis and low grip strength is highly positively correlated with dynapenia (factor loadings of 0.693 in osteoporosis, −0.583 in grip strength, and 0.581 in dynapenia), which is compatible with the results shown in Table 1.

In Table 3, we further correlated dynapenia with several variables including basic characteristics (age, BMI, WHR), grip strength, ASM/ht^2^, osteopenia/osteoporosis, DXA-defined body composition, appendicular and central fat distribution, status of glycemic control, and use of medication related to body weight alternation (insulin, sodium–glucose co-transporter 2 inhibitor, and glucagon-like peptide 1 receptor agonist). Again, dynapenia showed a persistent positive correlation with osteoporosis in T2DM females even after BMI adjustment (r = 0.325, *p* < 0.05). In contrast, grip strength correlated negatively with dynapenia (r = −0.594, *p* < 0.01).

In Table 4, we separated T2DM females to dynapenic and non-dynapenic groups for specific comparison among age, BMI, grip strength, body composition, and regional fat distribution. Similar to the findings observed in Table 3, there are significantly higher BMI (*p* = 0.036), lower grip strength (*p* < 0.001), and increased prevalence of osteoporosis (*p* = 0.025) in T2DM females with dynapenia. Other parameters have no significant difference between groups.

In Table 5, we further performed a multivariable stepwise logistic regression aiming to identify novel parameters in T2DM females that potentially served as predictors for the occurrence of dynapenia. According to the results of Table 3 and Table 4, we found that BMI and osteoporosis were associated with dynapenia. Therefore, several main parameters related to body components and osteoporosis were included in the analysis (age, BMI, osteoporosis, total bone mineral content, total lean mass, total fat mass, lumbar BMD, left femur BMD, and right femur BMD). The results showed BMI and osteoporosis can be used as predictors to assess the occurrence of dynapenia in T2DM females. Every 1 unit increase in BMI will elevate the probability of dynapenia by 23.4%. More importantly, once the elderly T2DM females were diagnosed with osteoporosis, the risk of dynapenia will increase dramatically by 388.3%.

## 4. Discussion

Our study was to investigate whether there are sex differences in the prevalence of metabolic musculoskeletal diseases in elderly patients with T2DM. The results of the study found that elderly T2DM females (median 68.8 years old) had a higher prevalence rate of dynapenia and dyna–osteoporosis than males. Factor analysis disclosed that a rise in BMI is negatively correlated with sarcopenia in one group of T2DM females, but having osteoporosis is positively correlated with the development of dynapenia in another group. In addition, there is a persistently positive correlation between dynapenia and osteoporosis in T2DM females even after BMI adjustment. Directly comparing between T2DM females with dynapenia and those without dynapenia, we further demonstrated dynapenia was meaningfully associated with not only lower grip strength but also higher BMI and elevated prevalence of osteoporosis. Moreover, we conducted a multivariable stepwise logistic regression analysis and revealed that BMI and osteoporosis were considered as reasonable predictors to evaluate the occurrence of dynapenia in T2DM females with odds ratios (95% CIs) of 1.234 (1.029–1.480) and 4.883 (1.352–17.630), respectively. Generally, our results highlight dynapenia as an important health issue particularly in osteoporotic or overweight T2DM females.

Our data revealed that the overall prevalence of sarcopenia in patients with T2DM is 17.5%, without a significant difference between the sexes (15% in females and 20.9% in males, *p* = 0.434), whereas the overall prevalence of dynapenia is 24.3%, with a higher rate in females (36.7% in females and 7% in males, *p* = 0.001). Previous studies have investigated gender-specific differences in sarcopenia. One study noted that due to differences in lean body mass and strength between men and women, a separate analysis by sex showed that 21.7% of men and 13.7% of women had sarcopenia [26]. Another study found a significant association between sarcopenia and gender, revealing that the prevalence of sarcopenia in male peritoneal dialysis patients was significantly higher than in females. This disparity may be attributed to reduced testosterone secretion, resulting in decreased muscle protein synthesis and increased muscle protein catabolism [27]. In patients with T2DM, we observed a similar trend, although it was not statistically significant, suggesting that males may have a higher risk for sarcopenia than females. The overall prevalence of sarcopenia seen in our study is in line with previous reports showing 18% pooled prevalence of sarcopenia in patients with T2DM [28]. Regarding the prevalence of dynapenia, a recent study pointed out diabetes as an important risk factor for dynapenia, and around 13.9% of T2DM subjects aged ≥ 65 years were diagnosed with dynapenia [29]. Another investigation found that 31.3% of postmenopausal women had dynapenia [30]. The variated prevalence of dynapenia observed in these results might be due to the differences in the cut-off value, associated comorbidity, and ethnicity of the population. However, it is worth mention that being female and having a BMI ≥ 25.0 kg/m^2^ were recognized to be factors associated with the risk of dynapenia [29], and postmenopausal women with a higher femoral neck T-score were less likely to have dynapenia [30], which are all compatible with our observations.

The pathogenesis of dynapenia is likely different from sarcopenia [14,15]. A longitudinal study investigating the dynamic changes in muscle mass and strength during aging disclosed the annualized decreases in muscle strength (measured by isokinetic leg muscle torque) were 2–5 times greater than the loss of muscle mass (quantified by thigh muscle cross-sectional area) [31]. Therefore, the loss of muscle strength cannot be merely explained by the loss of muscle mass during aging. Indeed, Clark et al. [15] had performed a comprehensive review and described several underpinned mechanisms involved in the development of dynapenia, including impairments of neural activation (decrease in supraspinal drive, motor unit recruitment and discharge rate) and/or reductions in muscle contractile quality (excitation-contraction uncoupling and muscle fiber type transformation). Notably, there is some evidence to point out that females might be vulnerable to having impaired inter-connection between the muscular and neural system due to higher intensity of diabetes associated neuropathic pain [32] and decreased motor unit discharge rates in aged females [33]. Moreover, the sex-specific difference in the decline rate of grip strength was also demonstrated by evaluating 48,070 individuals who are at least 50 years old and engaged in vigorous physical activity at least once a week. The decline in grip strength in males accelerates with each year of increase in age. In contrast, the grip strength in females was maintained from ages 50 to 55 years with subsequently accelerated decline, possibly due to menopausal transition [34]. This female-related rapid decline in grip strength after menopause might be more obvious in patients with diabetes [35]. Overall, all these factors possibly contribute to the higher prevalence of dynapenia in T2DM females, as shown in our study.

The independently significant association between obesity and dynapenia in patients with T2DM had been disclosed in a Japanese multi-institutional joint cross-sectional study [29]. A similar finding was also observed in our data that dynapenic T2DM females presented a higher BMI than non-dynapenic ones (Table 4). Moreover, an age-associated increase in intramuscular fat was demonstrated in a longitudinal follow-up study [31], and the infiltration of adipocytes into muscle fibers was considered to play a role in the development of dynapenia [15]. Therefore, in this study, we used DXA-defined arms’ and legs’ fat-to-lean ratio as surrogates for regional intramuscular fat infiltration to evaluate their associations with dynapenia (Table 4). However, both of them showed no significant differences between dynapenic and non-dynapenic females. Further large-scale prospective follow-up or well-designed animal or human studies might be required to support or refute this hypothesis.

Our finding that osteoporosis was highly associated with dynapenia in females with T2DM warrants further attention. Osteoporosis, an important metabolic bone disorder, has a high prevalence among older individuals, particularly in females [36]. Numerous studies have demonstrated that hyperglycemia significantly impairs normal bone metabolism, leading to compromised bone quality at the microarchitecture level [5]. Despite having higher average BMD due to higher overall body weight, individuals with T2DM face an increased risk of osteoporotic fractures. To better understand this paradox, researchers have explored trabecular bone quality using a noninvasive assessment called the trabecular bone score (TBS). Notably, the TBS is decreased in T2DM, even when the BMD remains relatively high [37]. Abnormal trabecular microarchitecture may contribute to the observed fractures at higher BMD levels in T2DM. Additionally, hyperglycemia and insulin resistance also play roles in poor skeletal healthy in T2DM due to mechanisms associated with the production of advanced glycation end products and oxidative stress [37]. Moreover, the poor bone quality, coupled with the risk of polyneuropathy-associated falls, contributes to the elevated prevalence of fragility fractures [38]. Consequently, dynapenia and osteoporosis are major contributors to fragility syndromes in older adults. Both conditions share common features related to age-related changes in body composition, chronic inflammation, and hormonal imbalances. Frailty syndromes, characterized by diminished responses to stress, can lead to a decline in physiological functions across various systems, increasing the risk of function loss, falls, and mortality [39]. Collectively, while the BMD remains relatively preserved in T2DM, deficits in other aspects of bone quality, combined with the increased prevalence of dynapenia and fragile syndromes, significantly elevate the fracture risk in elderly patients with T2DM.

Recent research have also highlighted associations between osteoporosis and dynapenia. For instance, one investigation on post-menopausal women showed women with a higher femoral neck T-score were less likely to have dynapenia [30]. Also, the association between osteoporosis and low grip strength was revealed in a Korean national cross-sectional study [40], emphasizing the importance of muscle function in bone health. Additionally, low Serum 25(OH)D was found as a risk factor for the incidence of dynapenia [41]. Hence, the interplay between dynapenia and impaired bone density should be noted, especially in the high-risk groups, such as elderly females with T2DM. This combination significantly impedes the quality of life and increases the risk of fragile fractures. Future research, employing multidisciplinary approaches, such as exercise, nutrition, and vitamin D supplementation, should explore interventions that simultaneously address muscle function and bone density in T2DM patients.

Several limitations in our study should be mentioned. First, this was an observational cross-sectional study, and it was impossible to define the causal relationship among the factors we evaluated. Second, participants were recruited from the out-patient department of a single medical center with fair ambulatory activity, which limits the generalizability of our results. Third, we did not evaluate the history of fragile fractures in our population. Therefore, the prevalence of osteoporosis calculated in this study may be underestimated since osteopenic subjects with fragile fractures were also diagnosed with osteoporosis. Lastly, further prospective studies will be necessary to clarify the impact of dynapenia on the physical quality of life and incidental falls in females with diabetes. To our knowledge, there are only few reports on dynapenia in elderly patients with diabetes, and this is the first study to concurrently evaluate dynapenia and osteoporosis in diabetes. Our results will prompt clinicians to pay more attention to identifying dynapenia in subjects with a high risk of fragility fractures such as females with diabetes and osteoporosis.

## 5. Conclusions

In conclusion, our results point out that T2DM females had a high prevalence rate of dynapenia, which commonly co-occurs with osteoporosis. Moreover, osteoporosis and BMI might serve as predictors for the risk of dynapenia in elderly females with T2DM.

## Figures and Tables

**Table 1 jcm-13-04590-t001:** Sex-specific difference on the occurrence or co-occurrence of sarcopenia, dynapenia, osteopenia, and osteoporosis in T2DM patients.

	Females	Males	*p* Value
	(*n* = 60)	(*n* = 43)	
Sarcopenia % (*n*)	15% (9)	20.9% (9)	0.434
Dynapenia % (*n*)	36.7% (22)	7% (3)	0.001 *
Osteopenia % (*n*)	40% (24)	44.2% (19)	0.671
Osteoporosis % (*n*)	28.3% (17)	16.3% (7)	0.154
Sarco–osteopenia % (*n*)	3.3% (2)	9.3% (4)	0.202
Sarco–osteoporosis % (*n*)	6.7% (4)	9.3% (4)	0.622
Dyna–osteopenia % (*n*)	10% (6)	2.3% (1)	0.127
Dyna–osteoporosis % (*n*)	16.7% (10)	2.3% (1)	0.020 *

Categorical variables were analyzed using the Chi-square test and are presented as percentages (number). * *p* < 0.05. Sarco–osteopenia refers to individuals with concurrent sarcopenia and osteopenia. Sarco–osteoporosis refers to individuals with concurrent sarcopenia and osteoporosis. Dyna–osteopenia refers to individuals with concurrent dynapenia and osteopenia. Dyna–osteoporosis refers to individuals with concurrent dynapenia and osteoporosis.

**Table 2 jcm-13-04590-t002:** Factor analysis in female T2DM patients.

	Factors	Female (*n* = 60)	
Items		Factor 1	Factor 2
Age		0.032	0.429
BMI		0.979	0.105
WHR		0.384	0.039
Android-to-gynoid fat ratio		0.296	0.227
Arms’ fat-to-lean ratio		0.518	0.005
Sarcopenia		−0.645	0.024
Dynapenia		0.257	0.581
Grip strength		0.241	−0.583
ASM/ht2		0.557	0.056
Osteoporosis		−0.199	0.693
Lumbar BMD		0.175	−0.499

Exploratory factor analysis was performed using principal axis factoring and presented with factor loadings. Abbreviations: BMI, body mass index; WHR, waist-to-hip ratio; ASM, appendicular skeletal muscle mass; ht, height; BMD, bone mineral density.

**Table 3 jcm-13-04590-t003:** The correlations of dynapenia with various variables in female T2DM patients (*n* = 60).

Spearman Correlation	Dynapenia	
	*r*	*r* Adjusted for BMI
BMI	0.273 *	N/A
Age	0.174	0.202
WHR	0.026	−0.129
Grip strength	−0.590 **	−0.594 **
ASM/ht2	0.204	0.062
Osteopenia	−0.198	−0.211
Osteoporosis	0.289 *	0.325 *
Total lean mass	0.058	−0.128
Total BMC	−0.085	−0.218
Total fat mass	0.176	−0.129
Arms’ fat-to-lean ratio	0.160	0.011
Legs’ fat-to-lean ratio	0.124	−0.011
Android-to-gynoid fat ratio	0.122	0.044
Lumbar BMD	−0.172	−0.189
Left femur BMD	−0.120	−0.151
Right femur BMD	−0.106	−0.116
HbA1c	0.061	−0.068
Use of insulin	0.094	0.012
Use of SGLT2i	−0.224	−0.198
Use of GLP1RA	0.007	−0.082

Data were analyzed with Spearman correlation and presented with correlation coefficients (*r* or *r* adjusted for BMI). Abbreviation: BMI, body mass index; WHR, waist to hip ratio; ASM, appendicular skeletal muscle mass; ht, height; BMC, bone mineral content; BMD, bone mineral density; HbA1c, glycated hemoglobin; SGLT2i, sodium–glucose co-transporter 2 inhibitor; GLP1RA, glucagon-like peptide 1 receptor agonist. * *p* < 0.05; ** *p* < 0.01.

**Table 4 jcm-13-04590-t004:** Age, BMI, body composition, and regional fat distribution between non-dynapenic and dynapenic T2DM females.

Female	Dynapenic (*n* = 22)	Non-Dynapenic (*n* = 38)	*p* Value
Age (years)	69.8 [65.8; 73.2]	68.4 [60.9; 71.6]	0.182
BMI (kg/m^2^)	25.8 [23.9; 28.8]	24.2 [21.5; 26.8]	0.036 *
WHR	0.95 [0.88; 1.01]	0.94 [0.90; 1.01]	0.834
Grip strength (kg)	15.4 [13.2; 16.3]	20.8 [18.0; 23.7]	<0.001 **
ASM/ht2 (kg/m^2^)	5.99 [5.74; 6.39]	5.88 [5.31; 6.27]	0.118
Osteoporosis % (*n*)	45.5% (10)	18.4% (7)	0.025 *
Total lean mass (kg)	34.7 [31.9; 38.3]	35.0 [30.9; 38.3]	0.656
Total BMC (kg)	1.87 [1.61; 2.31]	1.95 [1.74; 2.20]	0.514
Total fat mass (kg)	23.5 [17.3; 26.5]	18.4 [15.1; 24.9]	0.177
Arms’ fat-to-lean ratio	0.68 [0.48; 0.84]	0.60 [0.49; 0.73]	0.220
Legs’ fat-to-lean ratio	0.54 [0.41; 0.65]	0.45 [0.32; 0.64]	0.342
Android-to-gynoid fat ratio	0.69 [0.57; 0.80]	0.63 [0.55; 0.76]	0.349
Lumbar BMD (g/cm^2^)	1.00 [0.84; 1.18]	1.07 [0.95; 1.17]	0.187
Left femur BMD (g/cm^2^)	0.85 [0.69; 0.95]	0.88 [0.77; 0.94]	0.357
Right femur BMD (g/cm^2^)	0.88 [0.68; 0.93]	0.87 [0.77; 0.93]	0.416

Variables were analyzed using the Mann–Whitney U-test and are presented as median values and [quartiles]. Categorical variables were analyzed using the Chi-square test and are presented as percentages (number). Abbreviations: BMI, body mass index; WHR, waist to hip ratio; ASM, appendicular skeletal muscle mass; ht, height; BMC, bone mineral content; BMD, bone mineral density. * *p* < 0.05; ** *p* < 0.001.

**Table 5 jcm-13-04590-t005:** Analysis of predictors for dynapenia in female T2DM patients.

Female	Multivariable Stepwise Logistic Regression		
	OR	95% CI	*p*-Value
BMI	1.234	1.029–1.480	0.023
Osteoporosis	4.883	1.352–17.630	0.015

Variables include age, BMI, osteoporosis, total BMC, total lean mass, total fat mass, lumbar BMD, left femur BMD, right femur BMD were analyzed using multivariable stepwise logistic regression (Backward LR). Variables are removed in order as below: total fat mass, lumbar BMD, total BMC, left femur BMD, right femur BMD, age, total lean mass. Abbreviations: BMI, body mass index; BMC, bone mineral content; BMD, bone mineral density.

## Data Availability

The data are available in the article, or will be shared on reasonable request to the corresponding authors.

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
