# Peer review of "Osteoporosis as the Female-Specific Risk Factor for Dynapenia in Elderly Patients with Type 2 Diabetes"

_jcm, 2024, doi:10.3390/jcm13164590_

Round 1

Reviewer 1 Report

Comments and Suggestions for Authors

In the manuscript "Osteoporosis as the female-specific risk factor for dynapenia in 2 elderly patients with type 2 diabetes" the authors aimed at evaluating the relationship between osteoporosis and dynapenia in patients with T2D.

The topic is of interest, but several issues should be addressed.

1) in the introduction, authors should focus a bit more on the relationship between bone and muscular health and how these organs interact to maintain their homeostasis. Infact, scanty muscular mass impair bone metabolism and viceversa.

2) a recent review underlying the relationship between adipose tissue expansion and diabetes, specifically on how adipose tissue talks to the pancreas in diabetes pathogenesis has been published (10.1530/JOE-23-0313). I think this is important to point out that maintaining a good body composition is important also for prevention of diabetes and not only for its management

3) in the method, authors should clarify how they have chosen sample size, which was the primary aim and secondary aims

4) in the discussion more attention should be paid to the following points:

-why the authors think there is a sex difference in their results?

-what is novel in the relationship between dynapenia and osteoporosis?

5) a table summarizing the baseline characteristics of population would be important (males versus female), including anthropometric parameters, DEXA data, glucose and lipid metabolism, therapies

Comments on the Quality of English Language

Minor mistakes shuold be revised

Author Response

Osteoporosis as the female-specific risk factor for dynapenia in elderly patients with type 2 diabetes

RE: JCM-3006238 R1

Lu et al.

Reviewer #1:

In the manuscript "Osteoporosis as the female-specific risk factor for dynapenia in 2 elderly patients with type 2 diabetes" the authors aimed at evaluating the relationship between osteoporosis and dynapenia in patients with T2D.

The topic is of interest, but several issues should be addressed.

  • in the introduction, authors should focus a bit more on the relationship between bone and muscular health and how these organs interact to maintain their homeostasis. Infact, scanty muscular mass impair bone metabolism and viceversa.

Response:

Thank you for your suggestions.

We have elaborated and added on the description in Page 6, Line 89-95.

Thank you again for pointing these out.

  • a recent review underlying the relationship between adipose tissue expansion and diabetes, specifically on how adipose tissue talks to the pancreas in diabetes pathogenesis has been published (10.1530/JOE-23-0313). I think this is important to point out that maintaining a good body composition is important also for prevention of diabetes and not only for its management

Response:

Thank you for your suggestions.

We have elaborated and added on the description in Page 4, Line 60-65.

Thank you again for pointing these out.

  • in the method, authors should clarify how they have chosen sample size, which was the primary aim and secondary aims

Response:

Thank you for your suggestions.

We have elaborated and added on the description in Page 8, Line 114-119, and the  primary aim and secondary aims as following :

  1. Gender association between the development of dynapenia and osteoporosis
  2. Which parameters are crucial in the development of dynapenia in T2DM females

Thank you again for pointing these out.

4) in the discussion more attention should be paid to the following points:

-why the authors think there is a sex difference in their results?

 Response:

Thank you for your suggestions.

We have elaborated and added on the description in Page 18, Line 274-284.

Thank you again for pointing these out.

-what is novel in the relationship between dynapenia and osteoporosis?

 Response:

Thank you for your suggestions.

We have elaborated and added on the description in Page 21-23, Line 346-368.

Thank you again for pointing these out.

5) a table summarizing the baseline characteristics of population would be important (males versus female), including anthropometric parameters, DEXA data, glucose and lipid metabolism, therapies

 Response:

Thank you for your suggestions.

We have elaborated and added on the description in the Supplemental table 1 Basic characteristics, medical prescription, grip strength and DXA scan parameters in the T2DM patients.

Thank you again for pointing these out.

Comments on the Quality of English Language

Response:

Thank you for your suggestions.

We have revised the manuscript for English language quality and documented documentation.

Thank you again for pointing these out.

Minor mistakes should be revised

Response:

Thank you for your suggestions.

We have revised the manuscript for English language quality and documented documentation.

Thank you again for pointing these out.

Reviewer 2 Report

Comments and Suggestions for Authors

This cross-sectional study aimed to establish the association between osteoporosis and dynapenia, focusing on T2D females due to its high prevalence of osteoporosis and fragility. The authors included 103  T2D patients aged between 50 to 80 years. DXA and dominant hand grip strength measurements were performed to define body composition, osteoporosis, and dynapneia in a sex-specific manner. Both BMI and osteoporosis were identified as risk predictors for the development of dynapenia in female T2D patients. The authors concluded a female-specific high co-occurrence of osteoporosis and dynapenia in T2D patients. Moreover, having osteoporosis and increased BMI might boost the risk of dynapenia in elderly females with T2D. Compared to their previous report about this issue, the authors further calculated the height-adjusted appendicular skeletal muscle mass along with the dominant hand grip strength for diagnosing sarcopenia and dynapenia. This interesting study adds more data from an area that is not being studied enough, and given the growing age of the population, the topic is important.

The issues to be resolved:

1. Lines 90-94 should be at the end of the discussion

2.  Line 99,100 … please add information… center for recruitment…you sad outpatients center, name it, how many, the level of medical care in those hospitals (primary, secondary or tertiary centers)…

3. Lines 102,103, please add information about background antihyperglycemic therapy

4. Lines 195-208, please add numeric values in the Results section

Author Response

Osteoporosis as the female-specific risk factor for dynapenia in elderly patients with type 2 diabetes

RE: JCM-3006238 R1

Lu et al.

Reviewer #2:

This cross-sectional study aimed to establish the association between osteoporosis and dynapenia, focusing on T2D females due to its high prevalence of osteoporosis and fragility. The authors included 103  T2D patients aged between 50 to 80 years. DXA and dominant hand grip strength measurements were performed to define body composition, osteoporosis, and dynapneia in a sex-specific manner. Both BMI and osteoporosis were identified as risk predictors for the development of dynapenia in female T2D patients. The authors concluded a female-specific high co-occurrence of osteoporosis and dynapenia in T2D patients. Moreover, having osteoporosis and increased BMI might boost the risk of dynapenia in elderly females with T2D. Compared to their previous report about this issue, the authors further calculated the height-adjusted appendicular skeletal muscle mass along with the dominant hand grip strength for diagnosing sarcopenia and dynapenia. This interesting study adds more data from an area that is not being studied enough, and given the growing age of the population, the topic is important.

The issues to be resolved:

  1. Lines 90-94 should be at the end of the discussion

Response:

Thank you for your suggestions.

We have corrected the relevant paragraph to the end of the discussion in Page 23, Line 378-382.

Thank you again for pointing these out.

  1. Line 99,100 … please add information… center for recruitment…you sad outpatients center, name it, how many, the level of medical care in those hospitals (primary, secondary or tertiary centers)…

Response:

Thank you for your suggestions.

We have elaborated and added on the description in Page 8, Line 119-122.

Thank you again for pointing these out.

  1. Lines 102,103, please add information about background antihyperglycemic therapy

Response:

Thank you for your suggestions.

We have elaborated and added on the description (Page 8, Line 126) (Page 13, Line 192-195) in the Supplemental table 1 of basic characteristics, medical prescription in the T2DM patients.

Thank you again for pointing these out.

  1. Lines 195-208, please add numeric values in the Results section

Response:

Thank you for your suggestions.

We have elaborated and added on the description in Page14, Line 217-223.

Thank you again for pointing these out.

Reviewer 3 Report

Comments and Suggestions for Authors

This is an interesting obsrvational, single-center, cross-sectional study analyzing sex-differences in the association between dynapenia/osteoporosis in diabetic elderly patients with clear implications for improving clinical practice.

The most important concern about the paper is the absence of clear information about other confounding variables that are not clearly presented in the Tables and I would strongly encorauge the authors to improve their work by doing so.

1) There must appear a table with much more complete basal data comparing Males and females, including complete set of biochemical data (not only glusocse, BMI and HbAIC but also creatinine, glomerular filtration rate, albuminuria/proteinuria, diagnosis of CKD, body composition and treatments). It is specially important to include drugs not only related to diabetic treatment but also bone-related treatments, VD, or those related to falls such as psicotropic drugs, HTA, etc.). Potential Basal differences should be included in the multivariant analysis. 

Minor comments:

1) LOW BMI is associated with osteoporosis and fractures, wheres this results direct in opposite direction. Please discuss that. 

2) It is very surprisning that no differences were observed between Males and females in TAble 1 regarding osteopenia and osteoporosis. COuld you please try to explain that?

3) Minor typos should be corrected. 

Comments on the Quality of English Language

Fair, just some very minor editorial amamnendments shall be made

Author Response

Osteoporosis as the female-specific risk factor for dynapenia in elderly patients with type 2 diabetes

RE: JCM-3006238 R1

Lu et al.

Reviewer #3:

This is an interesting obsrvational, single-center, cross-sectional study analyzing sex-differences in the association between dynapenia/osteoporosis in diabetic elderly patients with clear implications for improving clinical practice.

The most important concern about the paper is the absence of clear information about other confounding variables that are not clearly presented in the Tables and I would strongly encorauge the authors to improve their work by doing so.

  • There must appear a table with much more complete basal data comparing Males and females, including complete set of biochemical data (not only glusocse, BMI and HbAIC but also creatinine, glomerular filtration rate, albuminuria/proteinuria, diagnosis of CKD, body composition and treatments). It is specially important to include drugs not only related to diabetic treatment but also bone-related treatments, VD, or those related to falls such as psicotropic drugs, HTA, etc.). Potential Basal differences should be included in the multivariant analysis. 

Response:

Thank you for your suggestions.

We have elaborated and added on the description (Page 8, Line 126) (Page 13, Line 192-195) in the Supplemental table 1 of basic characteristics, medical prescription, grip strength and DXA scan parameters in the T2DM patients.

Thank you again for pointing these out.

Minor comments:

  • LOW BMI is associated with osteoporosis and fractures, wheres this results direct in opposite direction. Please discuss that. 

Response:

Thank you for your suggestions.

Our study suggested there is a female-predominant association between the development of dynapenia and osteoporosis.

Table 2: In one group, the rise of BMI is negatively correlated with sarcopenia but not with dynapneia.

Table 3: dynapenia showed persistently positively correlated with osteoporosis in T2DM females even after BMI adjustment (r = 0.325, P < 0.05). In contrast, grip strength correlated negatively with dynapenia (r = -0.594, P < 0.01).

Table 4: Similar to the findings observed in Table 3, there are significantly higher BMI (P = 0.036), lower grip strength (P < 0.001) and increased prevalence of osteoporosis (P = 0.025) in T2DM females with dynapenia.

According to the results of Table 3 and Table 4, we found that BMI and osteoporosis were associated with dynapenia.

Table 5: The results showed BMI and osteoporosis can be used as predictors to assess the occurrence of dynapenia in T2DM females. Every 1 unit increase of BMI will elevate the probability of dynapenia by 23.4%.

Thank you again for pointing these out.

  • It is very surprisning that no differences were observed between Males and females in Table 1 regarding osteopenia and osteoporosis. Could you please try to explain that?

Response:

Thank you for your suggestions.

Although our results show that the osteopenia/osteoporosis between males and females does not reach a statistical difference, females tend to have a higher prevalence rate of osteoporosis. The lack of statistical difference may be because we mainly received cases that OPD patients basically still have normal mobility, so the actual prevalence of osteoporosis in females may be underestimated.

Thank you again for pointing these out.

  • Minor typos should be corrected. 

Response:

Thank you for your suggestions.

We have revised the manuscript for English language quality and documented documentation.

Thank you again for pointing these out.

Comments on the Quality of English Language

Response:

Thank you for your suggestions.

We have revised the manuscript for English language quality and documented documentation.

Thank you again for pointing these out.

Fair, just some very minor editorial amamnendments shall be made

Response:

Thank you for your suggestions.

Thank you again for pointing these out.

Round 2

Reviewer 2 Report

Comments and Suggestions for Authors

The authors corrected the manuscript according to the suggestions, I have no further unresolved issues. 

Reviewer 3 Report

Comments and Suggestions for Authors

Thanks for the review